# Comparison of the Properties of Natural Sorbents for the Calcium Looping Process

**DOI:** 10.3390/ma14030548

**Published:** 2021-01-24

**Authors:** Krzysztof Labus

**Affiliations:** Department of Applied Geology, Silesian University of Technology, 2 Akademicka St., 44-100 Gliwice, Poland; krzysztof.labus@polsl.pl

**Keywords:** TGA-DSC, calcium looping, CO_2_ capture, mineral carbonation, natural sorbents, carbonate rock, serpentine

## Abstract

Capturing CO_2_ from industrial processes may be one of the main ways to control global temperature increases. One of the proposed methods is the calcium looping technology (CaL). The aim of this research was to assess the sequestration capacity of selected carbonate rocks, serpentinite, and basalt using a TGA-DSC analysis, thus simulating the CaL process. The highest degrees of conversion were obtained for limestones, lower degrees were obtained for magnesite and serpentinite, and the lowest were obtained for basalt. The decrease in the conversion rate, along with the subsequent CaL cycles, was most intense for the sorbents with the highest values. Thermally pretreated limestone samples demonstrated different degrees of conversion, which were the highest for the calcium-carbonate-rich limestones. The cumulative carbonation of the pretreated samples was more than twice as low as that of the raw ones. The thermal pretreatment was effective for the examined rocks.

## 1. Introduction

The prevailing view today is that anthropogenic and geogenic greenhouse gases (GHGs) emitted into the atmosphere are the main cause of global warming. Energy production and use may be responsible for almost two-thirds of global greenhouse gas emissions (e.g., [1]). CO_2_ is considered to be the greenhouse gas with the greatest contribution to global warming, and global CO_2_ emissions are used as a clear indicator of global fossil energy consumption (e.g., [2]). Given the above assumptions, it can be concluded that capturing CO_2_ from industrial processes may be one of the main ways to control global temperature increases.

Mineral carbonation (MC) is considered to be one of the safest technologies for reducing CO_2_ emissions into the atmosphere and is used to capture and store CO_2_ in situ—in geological formations—or ex situ—as a potential solution for CO_2_ sequestration from smaller emitters where geological sequestration is not a viable option [3].

The advantage of mineral carbonation is the permanent storage of CO_2_ in the form of thermodynamically stable and environmentally friendly carbonates (e.g., [4,5,6]). This process is exothermic, and the raw materials for its operation are widely available (which is advantageous from an economic point of view) [7]. In the MC process, appropriately selected mineral substrates react with CO_2_ and form thermodynamically stable carbonates. This prevents emissions and ensures permanent CO_2_ sequestration [8,9]. One of proposed ex situ methods is the calcium looping technology—CaL [10].

Calcium looping (CaL) systems have been proposed as a less expensive method of CO_2_ capture for conventional power plants. In this process, a key role is played by calcium sorbent, which is used in alternating calcination and carbonation processes. The efficiency of the process varies depending on the properties of the sorbents used, which are expressed, inter alia, in the effects observed for the decreasing efficiency of gas capture with increasing number of CaL cycles. It is believed that this phenomenon is related to the reduction of the active surface of the sorbent due to sintering and, possibly, the decrease in the chemical activity resulting from the reaction with sulfur oxides competing with carbonation. The reaction in which sulfur compounds are involved is largely similar to carbonation; however, it is irreversible under CaL conditions. It takes place in pores of small dimensions, and its products are deposited on the sorbent surface, which, in turn, makes carbonation difficult. The environments of carbonation are meso- and micropores, and especially in the latter ones, rapid filling with reaction products can take place [11].

The aim of this work was to assess the sequestration capacity of selected rocks using a simultaneous TGA-DSC analysis, thus simulating the calcium looping process. Such a method is suitable for small samples (e.g., drill cuttings, rock fragments), which are easy to obtain even at a very early stage of the raw material deposit recognition. Moreover, such tests do not require extensive reactors, and, in a relatively quick and simple way, they allow for the characterization of the material or screening of samples in terms of their suitability for CaL.

The calcium looping process—shown in Figure 1—uses a reversible chemical reaction,
CaO + CO_2_ = CaCO_3_,(1)
between lime (CaO) and CO_2_, to capture CO_2_ from waste gas streams [10]. CO_2_ in the gas stream reacts with CaO in an exothermic carbonation reaction, forming calcium carbonate (CaCO_3_) at temperatures in the range of 600–700 °C. The CaCO_3_ from the carbonizer is then sent to a separate device called a calciner, where the calcination reaction takes place at a high temperature (around 900 °C). In these conditions, high-purity CO_2_ is released, which is suitable for transport to the sequestration site. The CaO produced is then sent back to the carbonator, closing the loop. Many researchers have proposed the oxy-combustion of coal in a calciner as a heat source for the calcination reaction [10,12]. The heat can be recovered from the exothermic carbonation reaction as well as from high-temperature gas and solid waste streams to generate electricity. As a result, CaL CO_2_ capture technology can be less energy intensive and more economical than the amine-based chemical absorption process.

The efficiency of CO_2_ capture (carbonation) by the sorbent and its regeneration (calcination) depends on the reaction kinetics, the sorbent grain size, its specific surface area, and the pore space characteristics. The cyclicity of the CaL process is accompanied by a decrease in the active surface of the sorbent particles due to the tight packing of CaO (regular system) in comparison to CaCO_3_ (trigonal system). Capture of CO_2_ by the CaO phase occurs in two stages. The first stage, in which the surface of the sorbent is covered with calcium carbonate, is characterized by a fast reaction rate and is strongly dependent on the partial pressure of CO_2_ [14,15]. The second stage is slow—the contact of CO_2_ with the sorbent depends on diffusion through the CaCO_3_ coating. Therefore, CaL installations should be based on the use of the first stage [16]. The capture process can be described by, among others, the shrinking core model (SCM), in which both stages are included, as shown in Figure 2. According to this model, the reaction proceeds at a narrow front that moves into the solid particle. The reactant is completely converted as the front passes by [17].

The conversion of sorbent can be expressed on the basis of mass change as follows:(2)$$X=\left(\frac{m-m_{0}}{m_{0}}\right)\cdot \frac{M_{CaO}}{M_{CO_{2}}}$$
where *m* is the sorbent mass at the time t, *m*_0_ is the initial mass of the sorbent, and *M_CaO_* and *M_CO_*_2_ are the molar masses of *CaO* and *CO*_2_.

According to [17], given the spherical shape of sorbent particles, the relationship of sorbent conversion with particle radius is given by the formula:(3)$$1-X=\left(\frac{volume\;of\;unreacted\;core}{total\;volume\;of\;particle}\right)=\frac{\frac{4}{3}\pi r_{c}^{3}}{\frac{4}{3}\pi R^{3}}={\left(\frac{r_{c}}{R}\right)}^{3}$$

Let the time for complete conversion of a particle be τ (s), and t is the time of the carbonation reaction (s). Then, in terms of the fractional time for the carbonation reaction, the conversion of the sorbent is given by:(4)$$X=1-{\left(\frac{r_{c}}{R}\right)}^{3}=\frac{t}{\tau }.$$

Thus, we obtain the general relationship of time with the radius and with the conversion, and the progression of the chemical reaction in terms of the fractional conversion becomes:(5)1−(1−X)13 =tτ,
while the progression of diffusion is:(6)1−3(1−X)23 +2(1−X)=tτ.

## 2. Materials and Methods

In this work, rock samples representing the listed lithological types were studied as potential sorbents in the CaL process (Table 1).

The rock samples were mechanically ground using a PM 100 CM ball mill (Retsch, Haan, Germany) to a fraction of less than 0.08 mm. CO_2_ sorption studies were carried out on the STA 449 F3 Jupiter (Netzsch, Selb, Germany) apparatus. Samples weighing about 15 mg were placed in an Al_2_O_3_ crucible. First, the mass change of the analyzed samples was measured (Table 1); this allowed the determination of the share of calcite (or dolomite) in carbonate rocks. Measurements were made in N_2_ atmosphere at a temperature of up to 1030 °C and a heating rate of 10 K/min. Next, the simulation of the CaL process was carried out using a temperature program (Figure 3), which consisted in heating (calcination) of the primary sample from a temperature of 40 °C to about 900 °C at a rate of 20 °C/min in a N_2_ atmosphere with a flow of 25 mL/min. Then, a 10 min isothermal section was introduced, and the temperature was lowered to about 650 °C. Then, the carbonation process was carried out, keeping the sample at this temperature for 10 min with the attached CO_2_ flow at the rate of 25 mL/min. The test was carried out in 10 cycles of alternating calcination and carbonation. During the measurements, changes in the mass of the sample over time (TGA) and heat flow (DSC) were recorded.

In this work, we the tested raw sorbents (unmodified) as well as selected limestone sorbents that were thermally modified by pre-heating the sample for one hour at 1000 °C in 100% N_2_ atmosphere.

## 3. Results

### 3.1. Raw Sorbents

#### 3.1.1. Dolomite

The simulation of the CaL process for dolomite showed an initial weight loss of the sorbent (calcination) amounting to 45.06% of its weight (Figure 4), which means that the sorbent was composed of almost pure CaMg(CO_3_)_2_. The dolomite derivatogram reveals that two reactions registered on the DSC curve as two adjacent endothermic effects at 740 and 870 °C. The first is responsible for the CO_2_ release from MgCO_3_, and the second from CaCO_3_; the sample mass losses are 23.59 and 21.47%, respectively. During carbonation, the CO_2_ capture was obtained, causing the sample mass changes in the range from 18.77% (first cycle) to 10.30% (10th cycle) (Figure 5). The gas capture efficiency decreased with increasing number of CaL cycles, which may be related to the decreasing active surface of the sorbent due to sintering.

It is worth noting that the carbonation process was not completed within the assumed time of 10 min. This is evident in the sample mass change (TGA) graph, which shows the rapid rise (reaction step—line 1) smoothly moving through the transition (line 2) towards the diffusion step (line 3). According to the SCM model, such effects are connected with the increasing thickness of the CaCO_3_ layer surrounding the unreacted CaO core. However, this last step was not fully completed, as shown by the line 3, which is tangent to a portion of the mass loss curve still deviates from the horizontal position. This means that in the case of dolomite sorbent, carbonation could be carried out for a longer time than assumed in the analyzed simulation. This is justified by the observation of the occurrence of a segment typical for the diffusion that was visible at the time of about 10 min after closing the CO_2_ flux to the reaction chamber of the furnace, when the atmosphere was not fully replaced with the protective gas (N_2_). In this case, the extension of the carbonation time may be associated with a potential reduction in the economics of the capture process, as the recorded increase in uptake by diffusion was only 0.29% in the first cycle, and in the next, it was about 1% of the sorbent sample mass (Figure 5). This issue would require further tests with extended carbonation time in order to calculate the amount of CO_2_ bound by the sorbent through the diffusion process.

#### 3.1.2. Saint Anne Mountain Limestone

The initial weight loss of the sorbent was observed (dehydration and calcination) to amount to 42.89% of its weight (Figure 6), which means that the sorbent was composed of almost pure calcium carbonate. Carbonation caused the sample mass changes in the range from 30.49% (first cycle) to 12.42% (10th cycle) (Figure 7).

The gas capture efficiency clearly decreased after the first cycle—to 24.28%—which may be related to the reduction in the active sorbent surface due to sintering. In subsequent cycles, this decline showed a downward trend. It should be noted that the reaction segment (visible as a steep mass increase) was shortened more and more in subsequent steps. At the same time, the elongation of the transition section was noticeable—this proves the increasing role of diffusion and confirms the sintering phenomenon. At the end of the assumed carbonation stage, a slightly inclined section of the sample mass increase was formed, which proves the diffusion process at that time (Figure 6). In addition, within about 5 min after closing the CO_2_ supply, small, unsystematic fluctuations in the mass of the sample are revealed (from −0.71% to +0.06%), followed by rapid calcination.

#### 3.1.3. Marl

A decrease in the weight of the sorbent was observed, amounting to 29.75%, of which about 4% corresponded to dehydration and dehydroxylation of clay minerals, and the remaining 25.73% corresponded to carbonate decomposition (Figure 8). Carbonation caused the sample mass changes within the range from 3.52% (first cycle) to 1.67% (10th cycle) (Figure 9). The efficiency of CO_2_ gas uptake decreased at a decreasing rate, which may be related to the reduction in the active surface of the sorbent due to sintering (Figure 12). The reaction segment was shortened in successive stages, and instead, the elongation of the transition section became noticeable. This proves the increasing role of diffusion and confirms the sintering phenomenon. Within about 5 min after closing the CO_2_ valve, slight fluctuations in the mass of the sample were revealed, with a clearly decreasing character with subsequent cycles. They were followed by rapid calcination.

#### 3.1.4. Nephelinite

The initial weight loss of the sorbent (poorly marked dehydration and calcination) was observed, amounting to 1.90% of its weight (Figure 10) and showing a small, potential share of carbonates (most likely filling cracks or voids formed during degassing of basaltic lava). During carbonation, the gas capture efficiency showed no systematic variability, and in most cycles, it ranged from 0.63 to 0.68%. In only the first cycle, the reaction section (steep mass increase fragment) was short, followed by the diffusion section. In subsequent stages, the reaction segments were higher. They were followed by slight fluctuations in mass, lasting until the end of the assumed carbonation stage.

#### 3.1.5. Magnesite

The significant initial weight loss of the sorbent amounted to 50.92% of its weight (Figure 11), which proves the negligible potential contributions of other carbonates. The gas capture efficiency showed a decreasing trend, burdened with a non-systematic component, and ranged from 0.85% (cycle 1) to 0.72% (cycle 10). As in the case of the previously described nephelinite, in only the first cycle, the reaction section was short, followed by the diffusion process. In the next stages, the reaction segments were higher. Fluctuations in mass lasted until the end of the carbonation stage, and were followed by relatively slow calcination.

#### 3.1.6. Serpentinite

The result of the DTA–TG analysis of serpentine sample is shown in Figure 12. Two endothermic peaks were found at temperatures of 623.1 and 701.9 °C due to the release of structural water. At a temperature of 834.1 °C, a large exothermic peak was visible, which represents the destruction of the serpentine crystalline structure and the formation of forsterite, enstatite, and clinoenstatite. For temperatures higher than 750 °C, the TGA analysis showed no significant weight variation. During carbonation, the sample mass changes reached from 1.42% (first cycle) to 0.95% (10th cycle). The efficiency of gas capture decreased at a decreasing rate. The reaction segment was shortened in successive stages, while the diffusion segment became more apparent, which confirms the sintering of the sample. About 5 min after the CO_2_ shut off, there was a slight decrease in the mass of the sample—typical for the first-order reaction—followed by a sharp but slight weight loss due to calcination.

### 3.2. Thermal Pretreatment of Sorbents

Calcium sorbents are characterized by a decreasing activity of capturing CO_2_ in each subsequent carbonation cycle. After approximately one hundred cycles, the asymptotically decreasing sorbent yield ranges from 7 to 15% for a 10 min carbonation time. It is believed that this is the result of changes in the sorbent’s morphology, during which its specific surface area decreases and the micropores disappear. In order to improve the activity of sorbents, the following enhancement techniques are used:Doping—aimed at postponing or avoiding sintering of sorbent in order to moderate sintering and abrasion of the sorbent (e.g., [18,19]). The effectiveness of doping depends on the concentration of the substrate used. Too low of a concentration will have no effect, while too high of a concentration may block the pores [12,20];Chemical treatment—to obtain a better sintering performance and more favorable pore area (e.g., [21,22]). Although the chemical treatment presents reactivity benefits, it has two drawbacks: the cost and availability of the acid and the marginal increase in CO_2_ uptake [23],Thermal pretreatment—to improve the conversion of CaO in long series of cycles and to stabilize the sorption capacity (e.g., [24,25]).

Research performed by Manovic and Anthony [25] and Manovic et al. [26] demonstrated that thermal pre-treatment could be an important method of improving conversion of CaO over long series of cycles. Such a phenomenon might be explained by a theory proposed by Lysikov et al. [27] and developed by Manovic and Anthony [25], according to which the repetitive carbonation/calcination cycles enhance the formation of a skeleton of interconnected CaO. This skeleton acts as the outer layer of the reactive CaO layer and stabilizes the sorption capacity.

The tests by Manovic and Anthony [25] showed that the particles were strongly sintered and that carbonization occurred on the surface of the solid particle. The pre-treatment resulted in the formation of an internal skeleton of the sorbent particles and protection of their integrity. When the sorbents are preheated, after the decomposition of CaCO_3_, ion diffusion continues and stabilizes the skeleton, which, due to its porous structure, is able to maintain significant carbonation (Figure 13).

Manovic and Anthony [20] studied the improvement of sorbent properties (Kelly Rock limestone) with 54.39% CaO content and loss on ignition (44.20%) through steam reactivation, thermal pre-treatment, and addition of calcium-aluminate-based pellets. The most promising results were obtained for powdered Kelly Rock limestone samples (<50 μm). For the preheating temperatures of 1000 °C, the sorbent was self-reactivated for the next 30 carbonation cycles. The highest conversions were obtained at 1000 °C, for which 49% conversion was achieved in the last cycle, with an average value of ∼45% for 30 cycles. Lower conversion values were obtained for samples pretreated at 1100 and 1200 °C. However, taking into account the self-reactivation effect, these results may also be promising, especially since heating the sorbent at these temperatures (in combination with granulation) gives better mechanical properties that could prevent its undesirable attrition. Assuming that the pretreatment time is an important parameter, a 6 h thermal activation study was also carried out on Kelly Rock powdered samples at temperatures of 900–1100 °C. It was confirmed that the pretreatment time influences the properties of the sorbent, and that shorter thermal treatment times can positively influence the effectiveness of the sorbent [25].

In this work, based on the results of the preliminary tests, three limestone samples from Stramberk (Czechia), Podlesie (Poland), and Butkov (Slovakia) (Table 1) were selected for further testing. They were characterized by high, medium, and the lowest weight loss, respectively. The tested sorbents were thermally modified by pre-heating the sample for one hour at 1000 °C in 100% N_2_ atmosphere. A further research cycle was carried out according to the procedure described earlier.

The effect of thermal pretreatment is presented for the example of limestone from Stramberk (Czechia) (Figure 14A,B). During the carbonation of the raw sample, CO_2_ capture was achieved, causing the sample mass to change within a range from 31.36% (first cycle) to 13.68% (10th cycle). The gas capture efficiency decreased with increasing number of CaL cycles. After the first cycle, the gas uptake efficiency decreased to 25.36%, which may be due to the reduction of the active sorbent surface due to sintering. It is noticeable that the reaction time (visible as a steep section of the mass increase) got shorter in subsequent cycles. Simultaneously with the reduction of the reaction section, the extension of the transition section was noticeable, which proves the increasing role of diffusion and confirms the phenomenon of sintering of the sample.

It is noteworthy that, within the assumed time of 10 min, the carbonation process was not completed. This is evident in the sample mass change graph, which shows a sharp rise (reaction stage) moving smoothly (transition stage) towards the diffusion stage. The latter, however, is not observed (no element approaching the horizontal line, mass growth curve). This means that in the case of the Stramberk limestone, carbonation could be carried out for a longer time than assumed in the analyzed simulation. The extension of the carbonation time in this case may be associated with a potential reduction in the economics of the capture process. This issue would require further tests with extended carbonation time in order to calculate the amount of CO_2_ bound by the sorbent through the diffusion process.

The comparison of the relative mass changes for Stramberk limestone without modification and thermally pretreated samples proves that the end of the reaction stage occurs at a similar temperature—around 644.5 °C (Figure 15). The calcination time varies, however, and is shorter for the unmodified sorbent, which is related to the lower content of CO_2_ blocked in this sample.

## 4. Discussion

The CaL simulations showed different degrees of conversion for the tested rock sorbents (Figure 16 and Figure 17). As expected, they were the highest for carbonate rocks (except for bituminous limestone); intermediate values were found for marl and bituminous limestone, and the values were lower by an order of magnitude for the remaining sorbents: magnesite and serpentinite. The lowest degree of conversion was determined for basalt.

The decrease in the conversion rate with subsequent CaL cycles was most pronounced for the sorbents with the highest values of this parameter, again with the exception of bituminous limestone (for which a significant decrease was noted between the first and second cycles; Figure 16). In the case of basalt, there were no significant changes in the conversion rate during the simulated cycles.

From the standpoint of the efficiency of the CaL process, an important indicator is the parameter called “cumulative carbonation relative to the initial mass of the sorbent”, which, for the 10 analyzed cycles (*n* = 10), could be defined as the degree of carbonation for *n* cycles.

It represents the multiplicity of the captured CO_2_ relative to the initial sorbent mass. In the course of the analyzed cycles, the cumulative carbonation shows regularities that are similar to the degree of conversion (Figure 18 and Figure 19). This parameter, exceeding 1.0, was also the highest for carbonate rocks (limestones: Stramberk—1.93, Saint Anne Mountain —1.81, Gorazdze—1.65, Podlesie—1.62, and dolomite of Olkusz—1.26); values that were lower by an order of magnitude were achieved for marl, bituminous limestone, and serpentine, and values lower by two orders of magnitude were achieved for magnesite and basalt.

The CaL simulations performed for the thermally pretreated samples also demonstrated different degrees of conversion for the tested rock sorbents (Table 2). As predicted, they were the highest for the Stramberk limestone sample, which was characterized by the highest proportion of calcium carbonate. The decrease in conversion with subsequent CaL cycles was most apparent for the sorbents with the highest values of this parameter.

The cumulative carbonation in the course of the analyzed cycles shows a pattern similar to that of the degree of conversion (Table 3); however, the value of this parameter was more than two times lower than for the raw samples of Stramberk (0.86) and Butkov (0.35). The greatest relative decrease was recorded for the Podlesie limestone, the cumulative carbonation of which decreased by more than three times (from 1.62 for raw sample to 0.5 after thermal treatment) despite the relatively high content of CaCO_3_.

The thermal pretreatment was not effective for the examined limestones, as also reported by Manovic et al. [20]; it is believed to be efficient for only some types of natural materials. It is likely that different types of limestone require different pretreatment conditions due to differences in impurities and internal structures [28]. However, this treatment has clear advantages: It is simple and relatively inexpensive compared to other techniques. On the other hand, it should be underlined that this would require additional energy to heat up the sorbent prior to its final use. This may result in a reduction in the power output of a CaL-equipped power generation system. Nonetheless, several studies proved that even if the pretreated limestone shows lower values of initial sorption capacity, this capacity increases over many cycles due to the softening of the hard skeleton. The disadvantage of this refining technique is that although the reactivity increases, the attrition of the particles significantly increases [29].

## 5. Summary and Conclusions

The suitability of selected rocks—limestone, dolomite, magnesite, marl, serpentinite, and basalt—was tested for the purpose of CO_2_ sequestration in the CaL process. TGA-DSC tests were carried out based on a temperature program designed for this purpose. The tests were performed in 10 cycles of alternating calcination and carbonation. During the measurements, changes in the mass of the sample over time (TGA) and heat flow (DSC) were recorded.

CaL simulations showed various degrees of conversion for the tested rock sorbents—the highest values were achieved for carbonate rocks (except for bituminous limestone), intermediate values were achieved for marl and bituminous limestone, lower ones were achieved for the remaining sorbents (magnesite and serpentinite), and the lowest were achieved for basalt.

The decrease in the conversion rate with subsequent CaL cycles was most intense for the sorbents with the highest values of this parameter. In the case of basalt, no significant changes in the conversion rate were observed. The decrease in gas capture efficiency with an increasing number of CaL cycles may be related to the decreasing active sorbent surface due to sintering.

The values of the parameter called “cumulative carbonation relative to the initial mass of the sorbent” corresponded to the multiplicity of the captured CO_2_ relative to the initial sorbent mass. This parameter, exceeding a value of 1.0, was the highest for carbonate rocks; it achieved lower values for marl, bituminous limestone, and serpentine, as well as—by two orders of magnitude—for magnesite and basalt.

In most of the analyzed samples, the carbonation process was not completed within the assumed time of 10 min. In practice, however, extending the carbonation time could reduce the economics of the capture process.

The simulations of the thermally pretreated samples also demonstrated different degrees of conversion for the tested rock sorbents, which were the highest for the calcium-carbonate-rich Stramberk limestone. The cumulative carbonation of the pretreated samples was more than two times lower than that of the raw ones. The largest relative decrease was recorded in the case of Podlesie limestone, the cumulative carbonation of which decreased by more than three times, despite the relatively high CaCO_3_ content.

## Figures and Tables

**Figure 1 materials-14-00548-f001:**
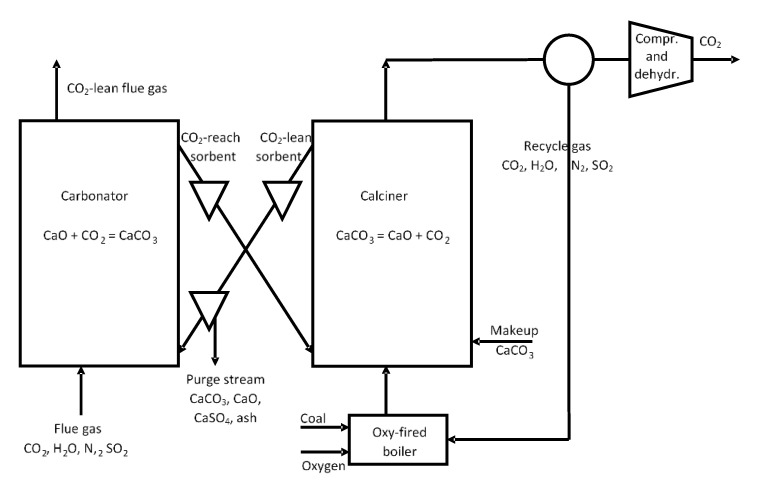
Diagram of the calcium looping (CaL) process for CO_2_ capture, according to [13].

**Figure 2 materials-14-00548-f002:**
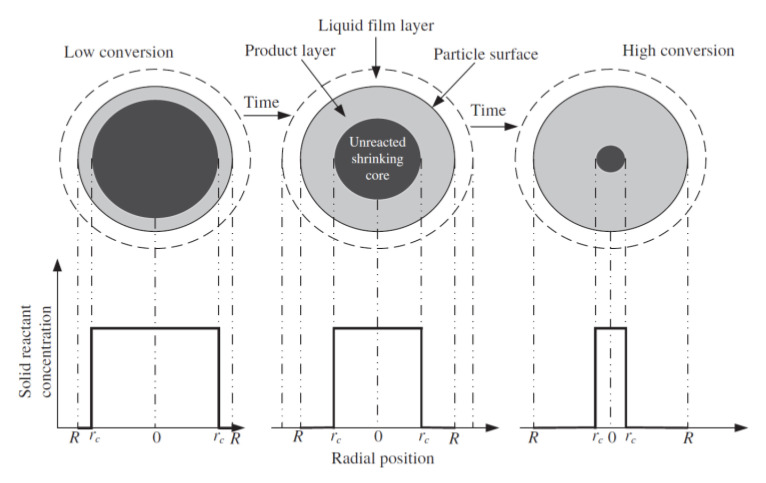
Schematic diagram of the shrinking core model (SCM), according to [17].

**Figure 3 materials-14-00548-f003:**
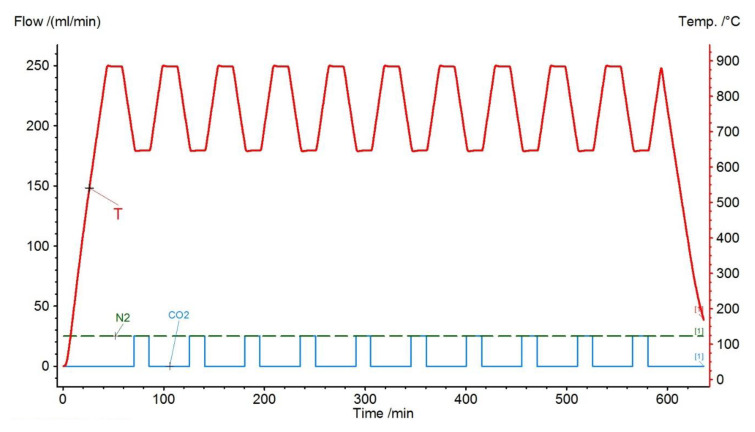
Example of a temperature program sequence (purple), and gas flow (green dashed line—N_2_, blue solid line—CO_2_).

**Figure 4 materials-14-00548-f004:**
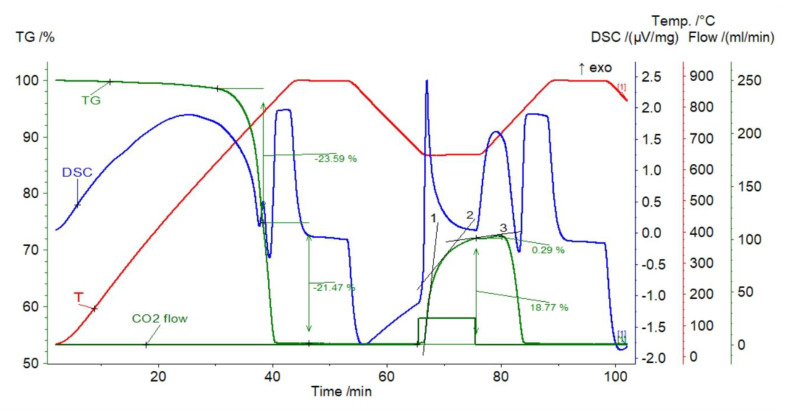
TGA and DSC curves for dolomite.

**Figure 5 materials-14-00548-f005:**
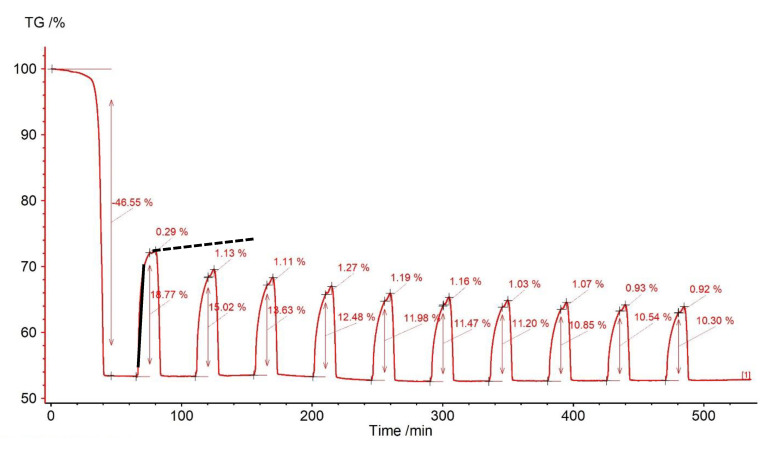
Relative mass changes for dolomite in the CaL process (the solid black line marks the section of the surface reaction, and the dashed line marks the section possibly corresponding to diffusion).

**Figure 6 materials-14-00548-f006:**
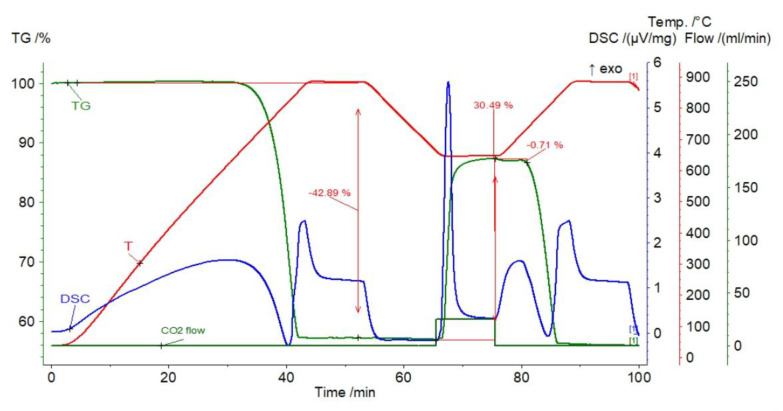
TGA and DSC curves for limestone from Saint Anne Mountain.

**Figure 7 materials-14-00548-f007:**
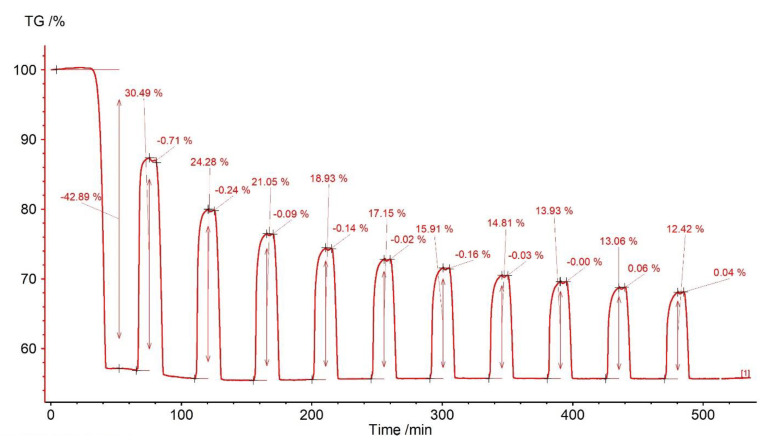
Relative mass changes for the limestone from Saint Anne Mountain.

**Figure 8 materials-14-00548-f008:**
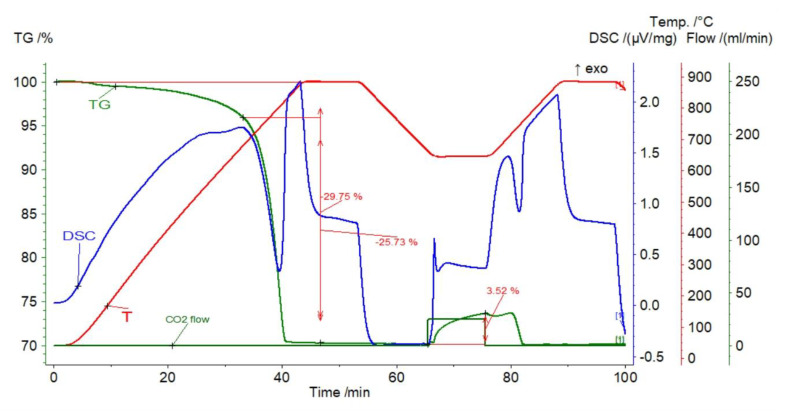
TGA and DSC curves for marl.

**Figure 9 materials-14-00548-f009:**
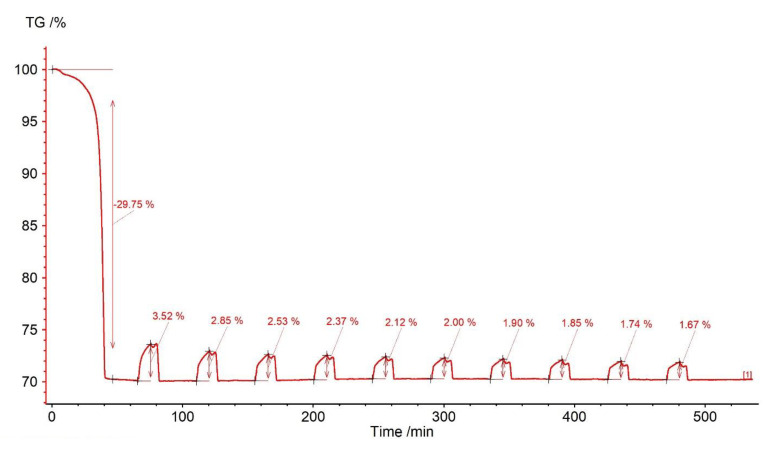
Relative mass changes for the marl.

**Figure 10 materials-14-00548-f010:**
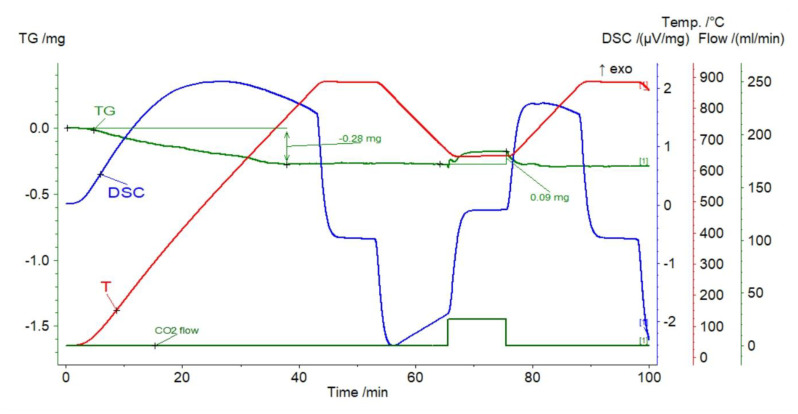
TGA and DSC curves for nephelinite.

**Figure 11 materials-14-00548-f011:**
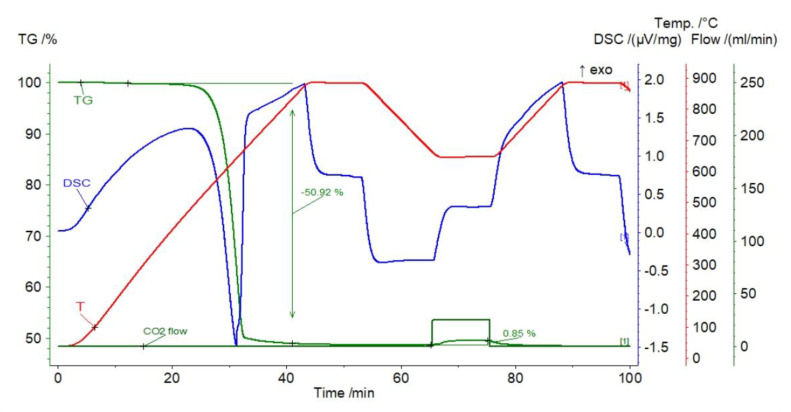
TGA and DSC curves for magnesite.

**Figure 12 materials-14-00548-f012:**
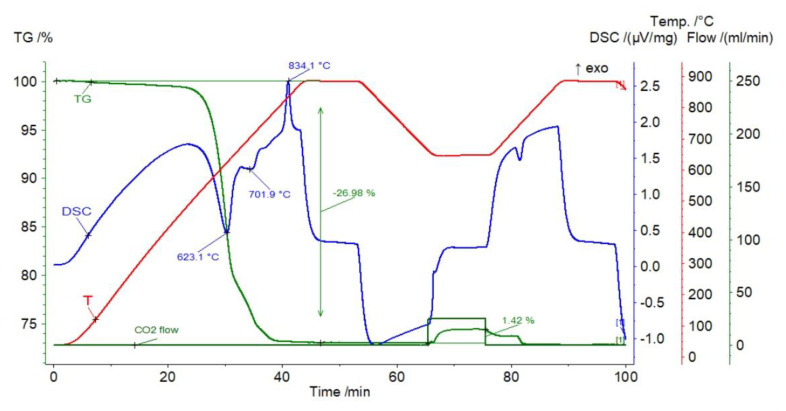
TGA and DSC curves for serpentinite.

**Figure 13 materials-14-00548-f013:**
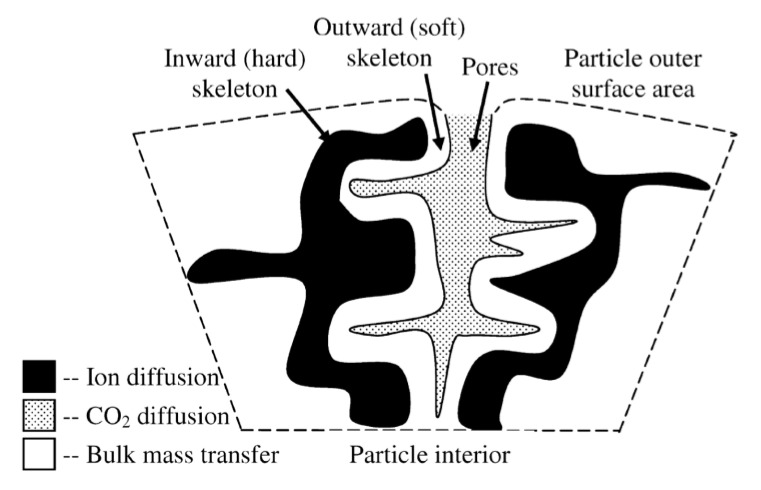
Schematic representation of the pore–skeleton model [25].

**Figure 14 materials-14-00548-f014:**
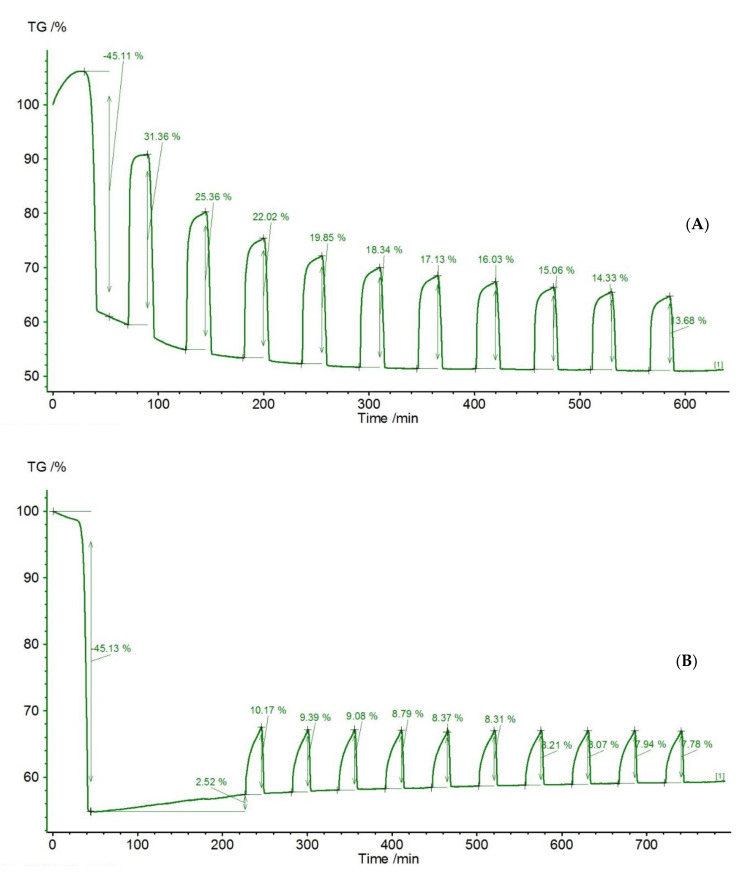
Relative mass changes for Stramberk limestone: (**A**) unmodified; (**B**) thermally pretreated for 1 h at 1000 °C.

**Figure 15 materials-14-00548-f015:**
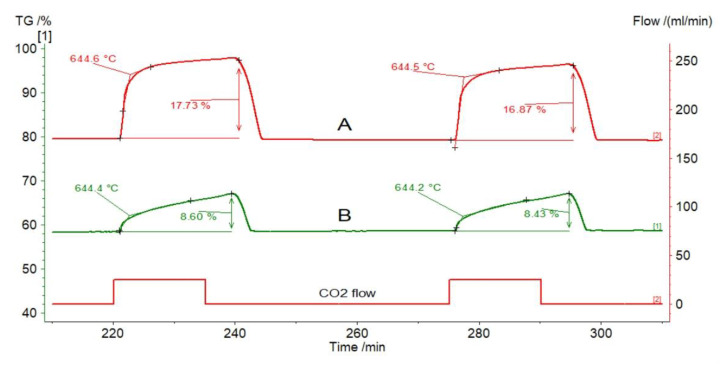
Relative mass changes for Stramberk limestone at the fifth and sixth calcium looping cycles: (**A**) without modification; (**B**) pretreated for 1 h at 1000 °C.

**Figure 16 materials-14-00548-f016:**
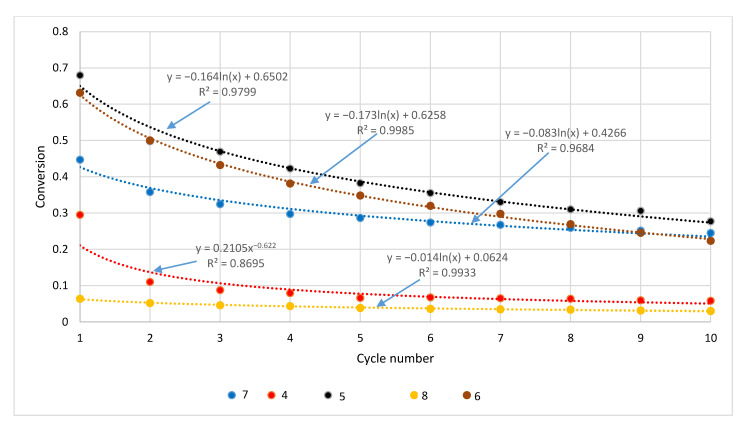
The conversion of carbonate rocks in the CaL process.

**Figure 17 materials-14-00548-f017:**
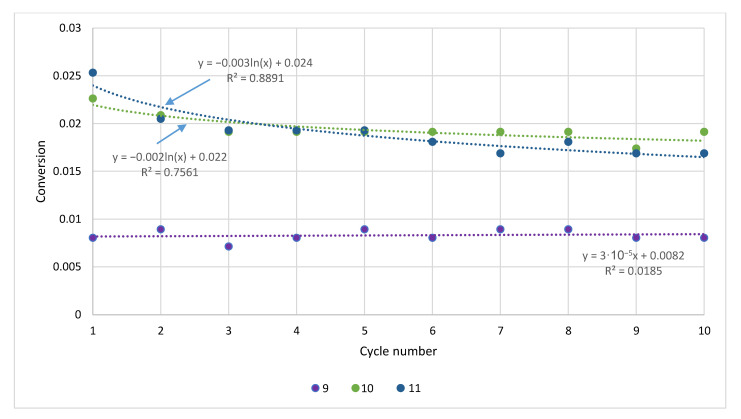
The conversion of magnesite and non-carbonate rocks in the CaL process.

**Figure 18 materials-14-00548-f018:**
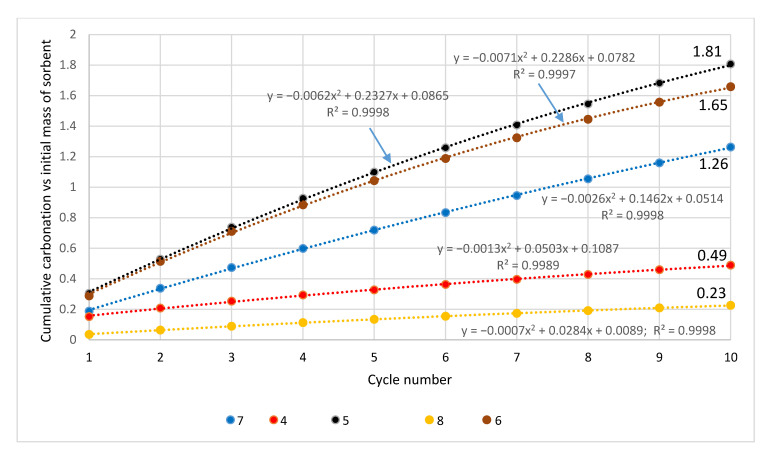
Cumulative carbonation relative to the initial mass of the sorbent—carbonate rocks.

**Figure 19 materials-14-00548-f019:**
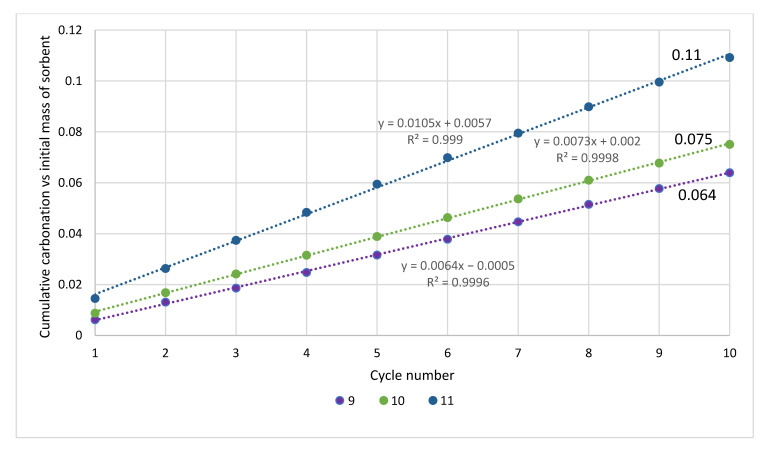
Cumulative carbonation relative to the initial mass of the sorbent—non-carbonate rocks.

**Table 1 materials-14-00548-t001:** Samples studied as potential sorbents in the CaL process.

Sample	Rock Type	Site/Age	Sample Mass [mg]	Mass Loss [%]	CaCO_3_ [%]
1	limestone	Stramberk(Czechia)/Jurassic	20.02	41.57	90.1
2	limestone	Podlesie (Poland)/Devonian	19.58	39.00	84.5
3	limestone	Butkov (Slovakia)/Cretaceous	20.32	31.81	68.9
4	bituminous limestone	Dębnik (Poland)/Devonian	14.72	34.51	78.5
5	limestone	Saint Anne Mountain (Poland)/Triassic	15.46	42.88	97.5
6	limestone	Gorazdze (Poland)/Triassic	13.9	42.09	95.7
7	dolomite	Olkusz (Poland)/Triassic	14.99	46.06	94.4 ^1^
8	marl	Cisownica (Poland)/Cretaceous	15.09	29.75	67.7
9	basalt (nefelinite)	Saint Anne Mountain (Poland)/Tertiary	14.55	1.92	-
10	magnesite	Braszowice (Poland)/Tertiary	14.91	50.91	97.5 ^2^
11	serpentinite	Jordanów (Poland)/older than UpperDevonian	14.46	26.97	-

^1^ % CaMg(CO_3_)_2_, ^2^ % MgCO_3_.

**Table 2 materials-14-00548-t002:** Conversion in raw and pretreated limestones.

Cycle	Sample					
	Štramberk		Podlesie		Butkov	
	90.1% CaCO_3_		84.5% CaCO_3_		68.9% CaCO_3_	
	Untreated	Pretreated	Untreated	Pretreated	Untreated	Pretreated
1	0.73	0.24	0.61	0.14	0.28	0.09
2	0.59	0.22	0.49	0.12	0.17	0.08
3	0.51	0.21	0.42	0.12	0.13	0.08
4	0.46	0.20	0.38	0.11	0.11	0.07
5	0.43	0.19	0.34	0.11	0.10	0.07
6	0.40	0.19	0.31	0.11	0.09	0.07
7	0.37	0.19	0.28	0.11	0.09	0.06
8	0.35	0.19	0.25	0.10	0.09	0.06
8	0.33	0.18	0.23	0.10	0.09	0.06
10	0.32	0.18	0.21	0.10	0.08	0.06

**Table 3 materials-14-00548-t003:** Cumulative carbonation relative to the initial weight of the sorbent for raw and pretreated limestones.

Cycle	Sample					
	Štramberk		Podlesie		Butkov	
	Untreated	Pretreated	Untreated	Pretreated	Untreated	Pretreated
1	0.31	0.10	0.28	0.06	0.14	0.05
2	0.57	0.20	0.50	0.12	0.23	0.09
3	0.79	0.29	0.70	0.17	0.29	0.13
4	0.99	0.37	0.87	0.22	0.35	0.16
5	1.17	0.46	1.03	0.27	0.40	0.20
6	1.34	0.54	1.17	0.32	0.45	0.23
7	1.50	0.62	1.30	0.37	0.50	0.26
8	1.65	0.70	1.41	0.41	0.54	0.29
8	1.80	0.78	1.52	0.46	0.58	0.32
10	1.93	0.86	1.62	0.50	0.62	0.35

## Data Availability

The data presented in this study are available on request from the corresponding author.

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
