# Peer review of "Comparison of the Properties of Natural Sorbents for the Calcium Looping Process"

_materials, 2021, doi:10.3390/ma14030548_

Round 1

Reviewer 1 Report

In this manuscript, the author presents results from a comparative study on applicability of some natural sorbents to capture CO2 from industrial processes using the calcium looping technique. Highest conversion rates were exhibited by carbonate rocks that also showed the highest decrease in the conversion rates at subsequent cycles. In addition, the author also investigated the effects of pretreating the samples thermally. The study is original and is significantly interesting. I therefore recommend accepting this work after the author carries out a revision addressing the following.

  1. Figure 2 describes the shrinking core model, but the relevant description in the text does not clearly state its connection with the two stages of capture of CO2.
  2. The flow rate in the figures is shown in ml/min, whereas the description in the text used ml/s. Even after conversion the numbers do not seem to match. Please check.
  3. The author uses a consistent color scheme to represent the TGA, DSC curves, temperature and gas flow data for Figures 4 and 6. This color scheme is however changed Figure 8 onwards. This unnecessary change can be a source of confusion. The author should use a consistent and uniform color scheme throughout the manuscript.

Reviewer 2 Report

Comments to Krzysztof Labus

Summary

The manuscript goes through various carbonate rock types experimentally evaluating their qualification for capturing CO2 from industrial processes. Furthermore, the author compares the CO2 capturing capacity of thermally pretreated limestone with that of the corresponding untreated one.

General comments

The topic of the study has current interest and is within the scope of the Materials journal. Moreover, the structure of the manuscript in terms of subsections is adequate for a scientific paper. In addition, the author provides enough description of the experiments and a careful report of the results. As for the introduction, otherwise fine, but the equations related to the shrinking core model seem a bit detached from the rest of the work.

The author demonstrates ample English skill and the figures and tables of the manuscript are of solid quality.

Specific comments

Line 8: Word order

Line 13: What parameter? If the reference is to the decrease in conversion rate, the sentence basically states the same thing twice.

Line 15: Expressions like two times lower render an ambiguous quality to a sentence.

Line 36: Punctuation

Line 41: Word order

Lines 67-68: Ill-formed sentence

Equation 2: What is the kinetic model (f(X)) of the reaction?

Equation 4: Where does this equation come from? Does it follow from equation 2?

Equation 5: Same questions as for equation 4

There is not enough detail in the mathematical description of the shrinking core model to make it helpful to the reader. It also seems that there is no further use of the equations in the manuscript; there is e.g. no attempt to estimate the parameters kc and kd for any of the rock materials. My suggestion would be either to provide some derivations, at least in the form of references, or to postpone the mathematical aspect of the shrinking core model for another article.

Line 107: …the listed lithological types were studied …

Lines 347-348: …parameter called “cumulative carbonation …”

Line 385: … was not effective …
